5

# MAX-DOAS retrieval of aerosol extinction properties in Madrid, Spain

Shanshan Wang<sup>1</sup>, Carlos A. Cuevas<sup>1</sup>, Udo Frieß<sup>2</sup>, Alfonso Saiz-Lopez<sup>1</sup>

<sup>1</sup>Department of Atmospheric Chemistry and Climate, Institute of Physical Chemistry Rocasolano, CSIC, Madrid, 28006, Spain <sup>2</sup>Institute of Environmental Physics, University of Heidelberg, Heidelberg, 69120, Germany

Correspondence to: Shanshan Wang (swang@iqfr.csic.es)

Abstract. Multi-Axis Differential Optical Absorption Spectroscopy (MAX-DOAS) measurements were performed in the urban environment of Madrid, Spain, from March to September in 2015. The  $O_4$  absorption in the UV was used to retrieve the aerosol extinction profile by an inversion algorithm. The results show a good agreement between the hourly retrieved aerosol optical depth (AOD) and the correlative Aerosol Robotic Network (AERONET) product, with a correlation coefficient of R = 0.87. Higher AODs are found in the summer season due to the more frequent occurrence of Saharan dust intrusions. The surface aerosol extinction coefficient as retrieved by the MAX-DOAS measurements was also compared to in situ PM<sub>2.5</sub> concentrations. The level of agreement between both measurements indicates that the MAX-DOAS retrieval

15 has the ability to characterize the extinction by particle near the surface. The retrieval algorithm was also used to study a case of severe dust intrusion on 12 May 2015. The capability of the MAX-DOAS retrieval to recognize the dust event including an elevated particle layer is investigated along with air mass back trajectory analysis.

# **1** Introduction

Atmospheric aerosols, with variable size ranging from a few nanometres to tens of micrometres in diameter, have influences on the atmospheric radiative budget, global climate change, local air quality and visibility, as well as directly or indirectly on human health (Seinfeld and Pandis, 2006; Kim and Ramanathan, 2008; Levy et al., 2013; Viana et al., 2014; Karanasiou et al., 2012). Moreover, the aerosol properties and vertical distribution vary greatly with geographical location. Therefore, it is important to obtain a comprehensive knowledge on the spatio-temporal distribution of aerosol characteristics in terms of chemical composition and physical properties. Measurements of aerosol optical and physical properties, including the

25 aerosol extinction coefficient (AEC), aerosol optical depth (AOD) and single scattering albedo (SSA), provide information for a better understanding of the aerosol's role in atmospheric processes. Additionally, more accurate vertically-resolved measurements of aerosol optical properties are still needed to further assess the aerosol environmental and radiative effects (IPCC, 2013).

Based on molecular UV-VIS light absorption, the Differential Optical Absorption Spectroscopy (DOAS) remote sensing 30 technique is an effective tool for air pollution measurements (Platt and Stutz, 2008). By observing scattered sunlight at

several elevations close to horizon and at zenith, Multi-axis DOAS (MAX-DOAS) is capable of retrieving information of the vertical distribution of numerous trace gases (Hönninger et al., 2004; Wittrock et al., 2004). Since the oxygen collision complexes  $O_4$  vertical profile is well known and nearly constant, the observed  $O_4$  absorption can serve as an indicator for the atmospheric distribution of photon paths (Wagner et al., 2004; Frieß et al., 2006). Therefore, the retrieved  $O_4$  differential

- slant column densities (DSCDs) at different elevations can provide information about the impact of aerosol scattering on photon paths. By combining measurements of the O<sub>4</sub> absorption with radiative transfer model simulations, ground-based MAX-DOAS has been used in previous studies to determine atmospheric aerosol vertical extinction profiles and optical depths (e.g. Irie et al., 2008, 2009; Li et al., 2010; Cl émer et al., 2010; Hartl and Wenig, 2013; Hendrick et al., 2014; Vlemmix et al., 2015; Frießet al., 2016).
- Madrid, the capital of Spain located in the south-west of Europe, is the third largest city in the European Union. The city of Madrid covers a total area of 604.3 km<sup>2</sup>. The residential population is almost 3.2 million, with a metropolitan area population of around 6.5 million. Due to the dense population and traffic, the city of Madrid has in recent years suffered severe levels of air pollution by nitrogen dioxide and occasionally by suspended particulates and ozone (Ayuntamiento de Madrid (AM), Madrid's Air Quality Plan 2011-2015, 2012). Air pollution episodes in the Madrid air basin are generally caused by local
- traffic emissions and domestic heating in winter (Plaza et al., 2011). The Madrid metropolitan area is bordered to the northnorthwest by a high mountain range about 40 km away from the city and to the northeast and east by lower mountainous terrain. This specific topography results in particular meteorological conditions and typical transport patterns that significantly influence air pollution dynamics in Madrid (Salvador et al., 2008). In addition, the city of Madrid also suffers from significant aerosol contributions from natural sources, e.g. occasional Saharan dust intrusions.
- To meet the current legislation targets on air quality, a new monitoring network of in situ instruments for the NO<sub>2</sub>, SO<sub>2</sub>, CO, O<sub>3</sub>, PM<sub>10</sub> and PM<sub>2.5</sub> is operational in Madrid since 2010. This monitoring network comprises 24 automatic measuring stations and two additional sampling points for PM<sub>2.5</sub> suspended particulates. The in situ PM<sub>2.5</sub> concentrations are measured at six stations as shown in Fig. 1, i.e. Esc. Aguirre, Casa de Campo, Quatro Caminos, Méndez Alvaro, Castellana and Pza. Castellana. Only a few studies have reported measurements of vertical aerosols optical properties in Madrid using Lidar systems over short periods of time (Molero et al., 2014; Fern ández et al., 2014).
- Here we present aerosol optical properties in the urban centre of Madrid, Spain, retrieved from MAX-DOAS observations of O<sub>4</sub> in the ultraviolet spectral region. This is the first time that measurements of aerosol extinction profiles in Madrid are reported for such a long period, allowing investigation of the temporal variation of aerosol optical properties in the urban area. We compare the AOD and surface aerosol extinction coefficient retrieved from MAX-DOAS with correlative sun
- photometer data and in situ PM<sub>2.5</sub> concentrations. Finally, we explore a case study of the aerosol optical properties during an intrusion of Sahara dust arriving in Madrid, which demonstrates the MAX-DOAS measurements as a useful monitoring tool to obtain vertically-revolved aerosol optical properties during dust episodes.

# 2 Measurements and Method

#### 2.1 MAX-DOAS instrument and setup

Our MAX-DOAS instrument consists of three main parts, i.e. the scanning system, the spectrometer-detector equipment and the computer as the control unit (e.g. Mahajan et al., 2012; Prados-Roman, et al., 2015). The scanning telescope was driven

- by a stepper motor to collect scattered sunlight from different elevation angles. The light is focused by a lens (F = 200 mm) with a diameter of 50.8 mm to a bundle of 15 individual quartz fibres. The light is fed by the fibre bundle to a Princeton Instruments SP500i spectrometer with a Princeton Instruments Pixis 400B CCD camera. The light is dispersed by a 600 grooves mm<sup>-1</sup> grating, resulting in a spectral window of 90 nm and spectral resolution of 0.5 nm FWHM. The computer is responsible to run the overall system and store the spectral data. During the spectra recording process, the offset was
- removed automatically. The signal of dark current was measured every night and subtracted from each spectrum according to the corresponding average exposure time. Depending on the intensity of the received scattered sunlight, the exposure time was adapted automatically between 0.1 and 1 s in order to optimize the total signal and avoid the saturation. Ground-based MAX-DOAS measurements were carried out from 15 March to 15 September 2015. The telescope scanning
- system was mounted on the top roof of a 25 m tall building at the main campus of the Spanish National Research Council (CSIC, 40.44 °N, 3.69 °W, 700 m a.s.l.) in Madrid, Spain (see Fig. 1). The spectra were recorded with an elevation angle sequence of -4 °, -2 °, 0 °, 1 °, 3 °, 5 °, 7 °, 10 °, 20 °, 30 ° and 90 ° for each scanning cycle. After each completed cycle, the grating was shifted between the UV and visible spectral regions, centred at 350 nm and 440 nm, respectively. The telescope was pointed approximately to the south-southeast (around 206 ° azimuth angle, red arrow in Fig. 1), towards the city centre of Madrid. This measurement site is located in the downtown area of Madrid and classified as a typical urban site, where the air
- quality is mainly impacted by nearby traffic emissions.

#### 2.2 Spectral analysis

The  $O_4$  DSCDs were derived from UV spectra covering 305 to 395 nm and only few data from visible spectra were available for inter-comparison. The QDOAS software developed by BIRA-IASB (<u>http://uv-vis.aeronomie.be/software/QDOAS/</u>) was applied to analyse  $O_4$  absorption in the interval between 339 and 387 nm, encompassing the three absorption bands at 343,

- 360 and 380 nm. Trace gas absorption cross sections for O<sub>4</sub> (Thalman and Volkamer, 2013), NO<sub>2</sub> (Vandaele et al., 1998), O<sub>3</sub> (Serdyuchenko et al., 2014) at 223 K and 243 K, BrO (Fleischmann et al., 2004) and HCHO (Meller and Moortgat et al., 2000) and a synthetic Ring spectrum (calculated by QDOAS) were included in the spectral fitting analysis. During each measurement cycle, the corresponding zenith spectrum was taken as a Fraunhofer reference spectrum for the lower viewing elevation angles. The relevant configuration for the spectral analysis is listed in Table 1. Afterwards, the O<sub>4</sub> DSCDs were
- introduced into the aerosol retrieval algorithm if relative errors yielded from QDOAS are less than 10%.

# 2.3 Other datasets

Cloud-screened AOD level 1.5 products from a CIMEL sun photometer situated on the roof of the Agencia Estatal de Meteorologia in Madrid, Spain (40.45 °N, 3.72 °W, 680 m a.s.l.) with a distance of 3.8 km from our measurements site (See Fig. 1) are compared to the column-integrated aerosol profile retrieved from MAX-DOAS measurements. The sun photometer is part of AERONET (Aerosol Robotic Network, http://aeronet.gsfc.nasa.gov/), which provides standards for instruments, calibration techniques, processing and data-distribution (Holben et al., 1998). The AERONET data is automatically cloud-screened if the direct view of the sun is blocked by clouds. Thus, the absence of AERONET data can serve as a temporal index for the presence of clouds and further help to filter the MAX-DOAS measurements (Smirnov et al., 2000). The AOD at wavelengths of 340, 380, 440, 500, 675, 870 and 1020 nm are available from the AERONET database at

10 the Madrid site.

Additionally, time series of PM<sub>2.5</sub> concentrations measured by in situ instruments were acquired from the air quality network of the Integral System of Air Quality Madrid City Council (<u>http://www.mambiente.munimadrid.es</u>) and used to compare with the surface aerosol extinction coefficient derived by the MAX-DOAS instrument. The six automatic measuring stations are located throughout the urban area of Madrid city (see Fig. 1).

Meteorological data, including temperature, relative humidity, wind speed and wind direction with a temporal resolution of 30 min, were obtained from the Adolfo Su árez Madrid–Barajas Airport meteorological site (40.47 °N, 3.56 °W, see Fig. 1) (<u>http://www.wunderground.com</u>). All data are normalized to 1-hour averages.

# 3 The HEIPRO aerosol retrieval algorithm

#### 3.1 Basic principle

- Since the zenith spectrum of each elevation sequence was taken as the reference, the DSCDs at lower elevations are sensitive to the trace gases in the lower troposphere. For the aerosol retrieval,  $O_4$  DSCDs from different elevations derived from the MAX-DOAS spectral analysis are fed into the aerosol inversion algorithm along with local atmospheric vertical pressure, temperature profiles and a suitable aerosol a priori profile. In this study, the HEIdelberg PROfile (HEIPRO, developed by IUP Heidelberg, Frieß et al., 2006; Frieß et al., 2011) retrieval algorithm, with the SCIATRAN radiative transfer scheme
- (Rozanov et al., 2005) as forward model, was used for the retrieval of aerosol vertical profiles. Based on the well-established optimal estimation method (OEM) (Rodgers, 2000), the HEIPRO algorithm determines the most probable atmospheric state x (aerosol extinction coefficient at a series of discrete altitude intervals) given a set of measurement  $y_m$  and a priori state vector  $x_a$ . The so-called maximum a posteriori (MAP) solution  $\hat{x}$  is approached by minimizing the cost function (Frießet al., 2006, 2011, 2016):

$$\chi^2(x) = [y - F(x, b)]^T S_{\epsilon}^{-1} [y - F(x, b)] + [x - x_a]^T S_a^{-1} [x - x_a],$$
 (1)

where the *m*-dimensional vector *x* defines the state of interest within the measured system. The radiative transfer model or forward function F(x, b) simulates the corresponding modelled value for the atmospheric state *x*, which is also dependent on the parameters *b* including temperature, pressure and aerosol profiles etc. *y* denotes the measurement vector, i.e. measured O<sub>4</sub> DSCDs at different elevation angles in the case of the aerosol profile retrieval. The a priori state vector  $x_a$  serves as an additional constraint.  $S_{\epsilon}$  and  $S_a$  denote the diagonal measurement covariance matrices representing the uncertainties in the

measurement and the a priori state, respectively.

The optimal state  $\hat{x}$  which minimises  $\chi^2$  can be found using the Levenberg-Marquardt method. The vertical resolution of the retrieval and the sensitivity of the retrieved state  $\hat{x}$  to the true atmospheric state x is quantified by the averaging kernel matrix  $A = \partial \hat{x} / \partial x$ . The retrieved profile  $\hat{x}$  can be represented as the true profile x, smoothed by the averaging kernel matrix A according to following Eq. (2):

$$\hat{x} = x_a + A(x - x_a) \tag{2}$$

Apart from the retrieval of aerosol properties, the HEIPRO algorithm can also yield the vertical profile of trace gases by combining the resulting aerosol extinction profiles, serving as forward modelling parameters, with measured DSCDs and a priori profiles of trace gases. However, this paper only focuses on the aerosol retrieval.

# 15 3.2 Aerosol inversion

In the forward radiative transfer model, atmospheric pressure and temperature profiles were adapted from the climatological data base employed in SCIATRAN, which contains monthly and latitudinal dependent vertical distribution of atmospheric trace gases, pressure and temperature. The surface albedo was set to 0.1 for the Madrid urban area. Aerosol extinction profiles from the surface up to 4 km height were retrieved on a vertical resolution of 100 m with a fixed temporal interval of

- 15 min. As to aerosol optical properties, the single scattering albedo and the asymmetry parameter for a Henyey-Greenstein phase function parameterization around 360 nm were set to 0.95 and 0.72 in the retrieval scheme, respectively. According to recently measured aerosol loadings in the Madrid downtown area, the annual average of surface  $PM_{2.5}$  and  $PM_{10}$  in 2014 were around 10.8 µg m<sup>-3</sup> and 19.4 µg m<sup>-3</sup>, respectively (Calidad del Aire Madrid 2014), and monthly averaged AOD at 440 nm ranged from 0.07 to 0.17. Based on these data, an a priori aerosol extinction profile with a surface extinction coefficient
- of 0.05 km<sup>-1</sup> and exponentially decreasing with scaling height of 1.5 km, was assumed with 100% error in retrieval. More details about the sensitivities of different a priori profiles will be discussed in Section 4.1. The consecutive measured O<sub>4</sub> DSCDs around the 360 nm band at different elevation angles serve as measurement vector for the HEIPRO algorithm. It is important to mention that the absorption of O<sub>4</sub> simulated with the absorption cross section from Hermans et al. (2003) was previously reported with an underestimation of 25% and suggested to be scaled by multiplying by
- 1.25 in several different aerosol retrieval schemes (Clémer et al., 2010; Großmann et al., 2013; Vlemmix et al., 2015). However, good agreement between modelled and measured O<sub>4</sub> DSCDs in the Arctic is also achieved without any correction for the same O<sub>4</sub> cross section (Frießet al., 2011). In fact, varied correction factors to the retrieved O<sub>4</sub> were applied depending

on the uncertainty introduced by the absolute value of the  $O_4$  absorption cross section (Zieger et al., 2011). Here, the new cross-section from Thalmann and Volkamer (2013) is adopted to retrieve  $O_4$ , therefore, we first need to include the appropriate scaling factor in the HEIPRO algorithm.

Consequently, a cloud-free day with low aerosol load (AOD < 0.15), i.e. 4 April 2015, was chosen to test different scaling factors for O<sub>4</sub> absorption, under which the O<sub>4</sub> DSCDs between 15° and 30° elevation are sensitive to variations of atmospheric temperature and pressure, as well as the aerosol optical properties (Frieß et al., 2006; Wagner et al., 2009, Cl émer et al., 2010). In the HEIPRO scheme, the scaling only takes effects on the modelled value by multiplying with the O<sub>4</sub> cross section. Since the Hermans et al., (2003) and Thalmann and Volkamer (2013) present different O<sub>4</sub> cross section values at 360 nm we test different scaling factors, 0.875, 1.0, 1.125, and 1.25, within HEIPRO in order to find the most appropriate

10 for the adopted Thalmann and Volkamer (2013) cross section.

In Fig. 2, the modelled  $O_4$  absorption with different scaling factors were compared to the measured  $O_4$  differential optical depth (DOD, Frieß et al., 2006) at elevation angles of 10°, 20° and 30°. It is obvious that the modelled  $O_4$  absorptions without scaling were systematically about 20% lower than the measured  $O_4$  at these elevations and even more during the morning periods. However, the modelled  $O_4$  absorption were overestimated with scaling factor of 1.25 around noon and even

at elevation angles lower than 10  $^{\circ}$ , which was also found by Irie et al. (2015). In the following aerosol retrieval at 360 nm, we assume a scaling factor of 1.2 as the optimal correction for the uncertainties from the newly available O<sub>4</sub> cross section, which is comparable to those previously reported for scaling O<sub>4</sub> from Hermans et al., (2003) (Wagner et al., 2009; Cl émer et al., 2010).

#### **4 Results and Discussion**

# 20 **4.1 Sensitivity to the a priori profile**

Since the information content of the measurement is usually too low to re-construct a full state vector, additional information concerning the atmospheric state is provided by an a priori state vector  $x_a$  with covariance matrix  $S_a$ . As an important input parameter, the a priori profile poses an additional constraint on the retrieved profile. Unfortunately, the impact of the a priori is substantial and there is no additional external information available which justifies the selection of one specific a priori

- 25 (Vlemmix et al., 2015). To investigate the impacts of a priori profile shape, aerosol retrievals were performed with four different a priori extinction profiles in the HEIPRO algorithm, i.e. linear, exponential, Boltzmann (smoothed box-shaped) and Gaussian distribution (peaking shape), as plotted in Fig. 3(a). The same cloud-free and low aerosol loading day of 4 April 2015 was chosen for the sensitivity study of different a priori profiles. Besides the input parameters in the forward model as mentioned in Section 3.2, other required observed geometry parameters were set according to the real measurements of a calor perith angle relative argin the angle.
- 30 measurements, e.g. solar zenith angle, relative azimuth angle.

5

Figure 3(b) shows the comparison between measured and simulated  $O_4$  absorptions resulting from the different a priori extinction profiles. When a Gaussian a priori was applied the simulated  $O_4$  absorption close to surface (elevation angle 1 degree) was much higher than the measured, and also resulted in underestimations at higher elevation angles. Except for the Gaussian a priori, there were only small differences in the modelled  $O_4$  absorptions among the other three a priori profiles. By estimating the degree of freedom for signal (DFS) and solution of cost function in the retrievals, the statistics indicates

- that better performance of larger DFS and smaller cost function were approached by utilizing an exponential a priori.
  Considering the fact that the mass concentration profiles of the different aerosol types usually decrease with altitude in the lower troposphere, whereas the background profile in the free troposphere remains constant with altitude, we have adopted the exponentially decreasing shaped extinction profile as the a priori. We then test the appropriate scale height of an
- 10 exponential a priori profile, which defines the decreases of aerosol concentration in vertical. The results show that the retrieved aerosol was constrained within lower altitudes by the algorithm if the scale height was set to be too small, whereas large scale height results in artefacts at higher altitudes. Therefore, a moderate scale height of 1.5 km, where the aerosol extinction coefficient decreased to the half of the surface value, was finally determined for the exponential a priori profile used in the aerosol retrieval.

# 15 4.2 Aerosol optical characteristics

To evaluate the performance of the MAX-DOAS retrieval, the AOD from the AERONET instrument at multiple wavelengths was interpolated to 360 nm using the Ångström coefficient ( $\alpha$ ) (see Eq. (3)):

$$4OD = \beta \times \lambda^{-\alpha}$$

(3)

- and then further averaged for hourly data series to compare with the MAX-DOAS retrieved AODs. Since AERONET data
  are automatically cloud cleared, the hourly MAX-DOAS retrieved AODs, which were calculated from the retrieved aerosol extinction profiles in HEIPRO algorithm, are normalized to the timetable of AERONET data. Figure 4 shows the time series of hourly AODs inferred from the sun photometer and MAX-DOAS measurements for several months. AODs retrieved from these two methods exhibit similar temporal trends. The occasional occurrences of high AODs (> 0.2) were attributed to the influence of long-range transport of windblown dust from the Saharan area, marked as grey hatched areas in Fig.4, which are further discussed as a case study in Sect. 4.4.
- The monthly averaged time series of AODs retrieved from MAX-DOAS and AERONET, shown in Fig. 5, exhibit higher aerosol loadings from June to September 2015. These higher AODs in the summer season are mainly due to long-range transport of Saharan dust, which occurs more frequently in summer than in winter (Sicard et al., 2011). Moreover, good agreement exists in both hourly and monthly retrieved AODs between MAX-DOAS and AERONET, as high correlation
- 30 coefficients of R=0.87 (Number of datapoints=618, AOD<sub>MAX-DOAS</sub>=0.6673\*AOD<sub>AERONET</sub>+0.0294) and R=0.96 (Number of datapoints=7, AOD<sub>MAX-DOAS</sub> =0.6253\*AOD<sub>AERONET</sub> +0.0362), respectively, were obtained by linear regression. However, these two datasets usually do not compare so well in case of higher aerosol loading, under which the absolute differences between the MAX-DOAS and AERONET are relatively larger (Fig. 4). The MAX-DOAS underestimation of the retrieved

AOD could be probably explained by the constraint that only aerosol extinctions below 4 km were considered in the algorithm, especially for conditions of dust events when the air mass can transport dust at higher altitudes. Furthermore, the exponentially decreasing a priori extinction profile poses strong constraints on aerosols at higher altitudes. Note also that the Saharan dust particles usually show the characteristic optical properties of single scattering albedo, asymmetry parameter and Ångström coefficient, which are different to those introduced in the basic scenario of HEIPRO retrieval (Gkikas et al.,

2013).

Generally, although a rather good correlation was obtained between MAX-DOAS retrieval and CIMEL instrument of AERONET, the deviation of these two data sets is still around 20%. This is possibly due to an inhomogeneous horizontal distribution of aerosols. Note that the AERONET site was located in the northwest away from the MAX-DOAS instrument,

whereas the telescope of MAX-DOAS points to a southern direction. As a consequence, the MAX-DOAS retrieved aerosol extinction profiles are representative for an average over the light paths in the lower troposphere over several kilometres up to 15 km in horizontal, which was roughly estimated based on the  $O_4$  SCD at 0 °elevation.

#### 4.3 Surface aerosol extinction

Owing to the absence of extinction coefficient measurements at ground surface and in vertical, another semi-quantitative

- way to validate the aerosol extinction coefficient retrieved by MAX-DOAS is to compare with particle mass concentrations. As indicated in Fig. 1, surface PM<sub>2.5</sub> concentrations from six in situ stations were averaged to represent the aerosol loading throughout the entire city, which are then compared to the MAX-DOAS retrieval of surface aerosol extinction coefficient, i.e. the bottom layer below 100 m of the aerosol extinction profile was considered as being representative for the surface extinction. Figure 6 shows the comparison between surface aerosol extinction coefficient retrieved from MAX-DOAS and 20 PM<sub>2.5</sub> concentration measured with in situ instrumentation.
- As shown in the upper panel of Fig. 6, a good agreement exists between the daily aerosol surface extinction coefficient from the MAX-DOAS retrievals and the ground  $PM_{2.5}$  concentration. This implies that the surface extinction coefficient obtained from MAX-DOAS retrieval truly reflects the amount of particles near the ground. The linear fitting for aerosol extinction coefficient and particle mass concentration yielded a correlation coefficient R of 0.89 in daily average and 0.64 in hourly
- average, respectively. Different explanations for the discrepancy in the hourly data include a) the regression analysis was based on the comparison between two different types of aerosol parameters, the aerosol extinction coefficient ranged basically with particle mass concentration, however, is also influenced by multiple factors including particles constituent, hygroscopicity and meteorological conditions (Zieger et al., 2011), b) the in situ PM<sub>2.5</sub> data were measured directly at or close to the surface whereas the aerosol surface extinction coefficient was extracted from the MAX-DOAS retrieved profile
- from ground to 100 m, averaged over a large horizontal distance. As a consequence, inhomogeneous distribution of particles both in vertical and horizontal direction decreases the consistency of these two data series (Frießet al., 2016). It also should be mentioned that the whole retrieved dataset were included in Fig. 6(c) without cloud filtering.

5

10

30

We then evaluate the impacts of air mass transport on the surface extinction coefficient and  $PM_{2.5}$ . The wind roses in Fig. 7(a) indicate that the AERONET AODs and retrieved surface extinction coefficient, as well as in situ  $PM_{2.5}$ , exhibit generally a similar dependence as a function of wind direction. As it can be seen in Fig. 7(a), combined high  $PM_{2.5}$  concentrations but low surface extinction conditions are present when the winds blow from S-SW-W direction. However, there was an inverse situation under northerly winds. In view of the specific geographic characteristics, i.e. the north of Madrid metropolitan is surrounded by mountains, the dispersion pathway of particles were significantly impacted by wind direction. Winds from N and S to SW components are the dominant from March to September in Madrid, see Fig. 7(b). By regression analysis of hourly  $PM_{2.5}$  and surface extinction coefficient for 16 different wind directions, correlation coefficients are found to be lower for north-eastern winds along with calm conditions, suggesting that the particles transported from the north-east show different extinction properties to those from other wind directions.

# 4.4 Case study of Saharan dust intrusion

Previous studies showed that the transport of Sahara dust resulted in high levels of aerosol pollution in the Iberian peninsula including the city of Madrid, while the Atlantic or polar air masses generally bring cleaner air, significantly reducing particulate matter levels (Querol et al., 2009; Di źz et al., 2012). During the period of MAX-DOAS measurements, several

- 15 Saharan dust intrusions affecting Madrid were reported by the Directorate-General of Environmental Quality and Assessment (Direcci on General de Calidad y Evaluaci on Ambiental) at the Ministry of Agriculture, Food and Environment (Ministerio de Agricultura, Alimentaci on y Medio Ambiente) (<u>http://www.magrama.gob.es/es/calidad-y-evaluacion-ambiental/temas/atmosfera-y-calidad-del-aire/calidad-del-aire/gestion/Prediccion\_episodios\_2015.aspx</u>), as also shown in Fig. 4. The diagnosis of Saharan dust intrusions uses an integrated approach based on data from back-trajectory analysis
- 20 (Hysplit model, <u>http://www.ready.noaa.gov</u>), simulated dust maps from the NRL (<u>http://www.nrlmry.navy.mil/aerosol/</u>), SKIRON (<u>http://forecast.uoa.gr/forecastnewinfo.php</u>) and BSCDREAM (<u>http://www.bsc.es/earth-sciences/mineral-dust-forecast-system/bsc-dream8b-forecast/</u>) models, and satellite images provided by the NASA SeaWiFS project, as described in Di źz et al., (2012).
- To investigate the capability of MAX-DOAS retrieval for dust intrusion, we use a case study focusing on the most severe dust intrusion with extremely high AOD up to 0.5, which occurred on 12 May 2015. Figure 8 (a) and (b) displayed the meteorological parameters during this episode, including temperature, relative humidity, wind direction and speed. It can be found the wind direction changed from northern to southern directions around 09:00 UTC on that day, while both  $PM_{2.5}$  and  $PM_{10}$  concentration, as well as AOD, increased to high levels that lasted for the following few hours. Together with high AOD, the arrival of the dry air mass is correlated with a relative humidity declined from 30% to 15%. Therefore, all these
- The Ångström coefficient, single scattering albedo and asymmetry parameters, as extracted from the AERONET product, are used to represent the characteristic optical properties of Saharan dust versus those of local aerosols. Figs. 8(e) and (f) depict the initial retrieved aerosol profile and the corrected after including in HEIPRO the characteristic aerosol optical properties

characteristics identified this day as a typical Saharan dust intrusion event.