# Peer review of "MAX-DOAS retrieval of aerosol extinction properties in Madrid, Spain"

_Atmospheric Measurement Techniques, 2016_

## Referee Comment (RC1) · Anonymous Referee #1 · 15 May 2016

The paper entitled "MAX-DOAS retrieval of aerosol extinction properties in Madrid,Spain"by Wang et al.present a half-year aerosol extinction properties retrieval based on MAX-DOAS measurement of O4 absorption in Madrid, Spain. The O4 DSCDs in the UV band was used to retrieve the aerosol extinction profile via the HEIPRO inversion algorithm. Not only the retrieved aerosol optical depth show anoverall good agreement with the correlative AERONET product, but also the surface aerosol extinction coefficients derived from MAX-DOAS measurement are comparable to in situ PM2.5 concentrations. The time series of AOD shows higher levels in summer season due to more frequent dust intrusions. Moreover, a case of severe dust intrusion was discussed for the performance of the MAX-DOAS retrieval to capture the dust event with an elevated particle layer. The potential causes of the systematical underestimation of MAX-DOAS retrieved AOD were discussed, especially during high aerosol

loading condition according to the case study. I have no major concerns that need to be addressed. The manuscript is generally well written, clearly presented and is recommended for publication in AMT after some minor corrections. Minor comments: P1, L9: UV-> UV spectral region P2, L27:Since no long-term aerosol extinction profile results were presented in the manuscript, it's better to describe with aerosol extinction properties, i.e. AOD, surface AEC. P3, L17&L22:The MAX-DOAS measurements are working in both UV and visible band, however, why did the authors only use the O4 absorptions in UV to retrieve the aerosol extinction profiles? Is it possible to show some inter-comparison of aerosol retrieval results between different wavelength bands? P6, L26:Why the sensitivity study of the a priori profile was tested with these four specific shapes? Is it restricted with the algorithm itself or any other reasons? Obviously, it is not realistic that Gaussian distribution as shown in Fig. 3(b). P7, L31: The comparison of AODs from MAX-DOAS and AERONET is not so well under high aerosol loading situation. If the correlation between them was improved if the dust days was excluded from the statistics? P8, L10: Please add one more sentence or proper reference for the estimation of light path based on O4 SCD at horizontal direction. P9, L32: Figs. -> Figure Figure 3: delete 'vertical' Figure 6: correlation plots -> linear regression plots

---

## Referee Comment (RC2) · Anonymous Referee #2 · 5 Aug 2016

The paper discusses the results of aerosol extinction vertical profile and AOD retrievals over Madrid, using MAX-DOAS $O_2$-$O_2$ ($O_4$) measurements in the UV for a period of several months in 2015. The results are validated with sun photometer and PM surface data. A Saharan dust event is analysed in detail.

The paper is clearly written. The methods used are well-known and described in sufficient detail. The results are interesting, and are relevant to using MAX-DOAS for aerosol detection and profiling.

The main comments are the following:

1. As discussed in the paper, esp. in sect. 4.1, the a priori aerosol profile is very important to get a good profile retrieval, and a good AOD value from the retrieval. It appears that an exponential profile works well for urban aerosols. However, this

exponential profile type is not realistic for a desert dust plume, which is an elevated layer of aerosols, including aerosols at altitudes above 4 km. This means that for desert dust events, like the one discussed in Sect. 4.4, another a priori profile should be used in order to get meaningful retrievals from MAX-DOAS retrievals. Please discuss this point, and extend the work of Sect. 4.4 for a Gaussian plume profile of an elevated dust layer.

2. It would be better to use an aerosol lidar for validation of MAX-DOAS aerosol extinction profiles. The AOD gives only the total column, and the surface concentration is only one point of the profile. Please discuss. Are there no lidars or ceilometers available in Madrid?

3. Use of satellite data, as mentioned on p. 9, l. 22: Daily satellite images of desert dust, visualized with the Absorbing Aerosol Index (AAI) are available from OMI and GOME-2 satellite data. See e.g. the TEMIS website: www.temis.nl, and go to Aerosol Index. These satellite images may help to select desert dust plumes and other elevated plumes of absorbing aerosols.

Minor comments including textual corrections are:

p. 1, l. 15: by particle > by aerosol particles

p. 1, l. 26: of the aerosol's role > of the role of aerosols

p. 2, l. 20: the NO2 > NO2

p. 2, l. 24: vertical aerosols > vertical aerosol

p. 2, l. 28: please specify the period. It is a matter of taste, but a few months is not long for a meteorological time series.

p. 3, l. 9: responsible to run > running (storing); which offset is meant?

p. 3, l. 12: the saturation > saturation

p. 3, l. 18: azimuth angle w.r.t. North

p. 3, l. 30: reformulate: . . . if the relative errors as found from QDOAS were less than 10 %.

p. 4, l. 28: measurements y_m

p. 4, l. 30 ff: explain chi square. Please use bold type for vectors and matrices. Please explain how y_m is linked to the vector y.

p. 5, l. 27: consecutively

p. 6, l. 6: Thalmann > Thalman. This occurs at more places in the paper.

p. 6, l. 24: a priori > a priori profile (this occurs at more places).

p. 7, l. 10: in vertical > in the vertical (this occurs at more places).

p. 7, l. 29: monthly > daily ?

p. 7, l. 29 – 31: please write out the hourly, daily and monthly correlations more clearly, in separate equations, as important results of this paper.

p. 8, l. 8: where does the 20 % come from?

p. 8, l. 14: the ground surface, the vertical

p. 8, l. 25: include:

p. 9, l. 5: metropolitan area

p. 9, l. 24: intrusions

p. 9, l. 25: display

p. 9, l. 27: found that

p. 9, l. 33: retrieved aerosol profiles and corrected profiles

p. 10, l. 16: remove: during day. Please make clear that Fig. 9 contains model AOD fields.

p. 10, l. 24: a cross-section scaling factor

p. 10, l. 28: exponential a priori profile

p. 14, l. 4: Stutz

p. 15, l. 2: quality

Figures:

Captions of Fig. 6 and Fig. 7: extinction > aerosol extinction

Fig. 1: please give the lat/lon coordinates as well.

Fig. 2: please order the lines according to increasing scaling factor. Please use a different color for the line with 1.25, since that color is too similar to red (line with 1.0). Please also show the SZA .

Fig. 3: In b the two red colors are too similar.

Fig. 4: please add a scatter plot to better see the ratio of MAX-DOAS AOD to Aeronet AOD.

Fig. 5: The figure contains a lot of double information. Please consider omitting figure (a).

Fig. 6: please indicate in the legend whether hourly or daily data is shown.

---

## Author Comment (AC2) · 30 Aug 2016

The paper discusses the results of aerosol extinction vertical profile and AOD retrievals over Madrid, using MAX-DOAS O2-O2 (O4) measurements in the UV for a period of several months in 2015. The results are validated with sun photometer and PM surface data. A Saharan dust event is analysed in detail.

The paper is clearly written. The methods used are well-known and described in sufficient detail. The results are interesting, and are relevant to using MAX-DOAS for aerosol detection and profiling.

The main comments are the following:

1. As discussed in the paper, esp. in sect. 4.1, the a priori aerosol profile is very important to get a good profile retrieval, and a good AOD value from the retrieval. It appears that an exponential profile works well for urban aerosols. However, this exponential profile type is not realistic for a desert dust plume, which is an elevated layer of aerosols, including aerosols at altitudes above 4 km. This means that for desert dust events, like the one discussed in Sect. 4.4, another a priori profile should be used in order to get meaningful retrievals from MAX-DOAS retrievals. Please discuss this point, and extend the work of Sect. 4.4 for a Gaussian plume profile of an elevated dust layer.

R: As it can be seen in the Fig.4 (a), the Gaussian profile was distributed as a peaking shape, which was parameterized in the algorithm with: a) the aerosol extinction coefficient of the peak; b) the altitude of the peak. Here, we have selected the Gaussian profile as a priori to retrieve the aerosol extinction during the dusty day. Because the bottom of the Gaussian profile close to the ground surface is extremely low aerosol extinction coefficient, it obviously deviates from the realistic condition. Consequently, the modelled O4 optical depths deviate from the measurements. During dust days, there are considerable aerosol loadings within low altitude, which means that this simplified Gaussian a priori is not suitable for the dust layer retrieval.

[Figure]

2. It would be better to use an aerosol lidar for validation of MAX-DOAS aerosol extinction profiles. The AOD gives only the total column, and the surface concentration is only one point of the profile. Please discuss. Are there no lidars or ceilometers available in Madrid?

R: Unfortunately, no lidar observation was operated during the MAX-DOAS measurements. We have tried to contact other research groups during the preparation of this paper, however, no

lidar or ceilometers data are available in Madrid. Considering the importance of lidar and ceilometers data to validate MAX-DOAS aerosol profile retrieval, we are planning to gather some other groups to perform a simultaneous measurement combining all the potential instruments in the coming future, e.g. lidar, MAX-DOAS, ceilometers.

3. Use of satellite data, as mentioned on p. 9, l. 22: Daily satellite images of desert dust, visualized with the Absorbing Aerosol Index (AAI) are available from OMI and GOME-2 satellite data. See e.g. the TEMIS website: www.temis.nl, and go to Aerosol Index. These satellite images may help to select desert dust plumes and other elevated plumes of absorbing aerosols.

R:Thanks a lot for the information. We are preparing another paper with the topic of desert dust intrusions in Madrid. These helpful data archives will be used to distinguish the desert dust plumes and other elevated plumes.

*Minor comments including textual corrections are:*

*p. 1, l. 15: by particle > by aerosol particles*

R: We have added the 'aerosol' in the sentence.

*p. 1, l. 26: of the aerosol's role > of the role of aerosols*

R: We have corrected it.

*p. 2, l. 20: the NO2 > NO2*

R: We have corrected it.

*p. 2, l. 24: vertical aerosols > vertical aerosol*

R: It has been corrected.

*p. 2, l. 28: please specify the period. It is a matter of taste, but a few months is not long for a meteorological time series.*

R: We have changed this sentence in the revised manuscript and specified the period.

*p. 3, l. 9: responsible to run > running (storing); which offset is meant?*

R:These two words have been changed in the revised manuscript.

*p. 3, l. 12: the saturation > saturation*

R:It has been corrected.

*p. 10, l. 16: remove: during day. Please make clear that Fig. 9 contains model AOD fields.*

R:'during day' has been deleted. It is now indicated in the caption of Fig. 9 that the shown AOD fields are modelled.

*p. 10, l. 24: a cross-section scaling factor*

R:'cross-section' has been added.

*p. 10, l. 28: exponential a priori profile*

R:'profile' has been added.

*p. 14, l. 4: Stutz*

R:It has been corrected.

*p. 15, l. 2: quality*

R:It has been corrected.

*Figures:*

*Captions of Fig. 6 and Fig. 7: extinction > aerosol extinction*

R: We have modified them.

*Fig. 1: please give the lat/lon coordinates as well.*

R: Because the different measurement sites are all located within the small area of urban Madrid. It's difficult to indicate the area of the map with lat/lon data. However, the lat/lon coordinates of MAX-DOAS and AERONET can be found in the text of manuscript, and we also have added them in the Fig. 1 itself. The detailed locations of other PM in-situ sites can be referred to *Ayuntamiento de Madrid (AM): Madrid's Air Quality Plan 2011–2015. General Directorate of Sustainability, Government Division of Environment, Safety and Mobility, Madrid City Council, 2012*.

*Fig. 2: please order the lines according to increasing scaling factor. Please use a different color for the line with 1.25, since that color is too similar to red (line with 1.0). Please also show the SZA .*

R: Figure 2 has been re-plotted with new legend and indicated with SZA as well.

*Fig. 3: In b the two red colors are too similar.*

R: In order to distinguish different a priori, the color indexes have been updated.

*Fig. 4: please add a scatter plot to better see the ratio of MAX-DOAS AOD to Aeronet AOD.*

R: It has been shown as Fig. 4(b).

*Fig. 5: The figure contains a lot of double information. Please consider omitting figure (a).*

R: Both of the Fig. 5(a) and (b) contain the monthly averaged values of AOD from AERONET and MAX-DOAS. However, more statistical parameters that median, $1^{st}/3^{rd}$ quartiles, and $5^{th}/95^{th}$ percentiles are shown in Fig. 5(a), whereas the standard deviations of monthly averaged AODs were indicated in Fig. 5(b). So we prefer to keep both of them in the revised manuscript.

*Fig. 6: please indicate in the legend whether hourly or daily data is shown.*

R:We have indicated the hourly and daily data with different legends in the Fig. 6(b) and (c), respectively.

---

## Author Comment (AC1)

*The paper entitled "MAX-DOAS retrieval of aerosol extinction properties in Madrid, Spain" by Wang et al. present a half-year aerosol extinction properties retrieval based on MAX-DOAS measurement of O4 absorption in Madrid, Spain. The O4 DSCDs in the UV band was used to retrieve the aerosol extinction profile via the HEIPRO inversion algorithm. Not only the retrieved aerosol optical depth show an overall good agreement with the correlative AERONET product, but also the surface aerosol extinction coefficients derived from MAX-DOAS measurement are comparable to in situ PM2.5 concentrations. The time series of AOD shows higher levels in summer season due to more frequent dust intrusions. Moreover, a case of severe dust intrusion was discussed for the performance of the MAX-DOAS retrieval to capture the dust event with an elevated particle layer. The potential causes of the systematical underestimation of MAX-DOAS retrieved AOD were discussed, especially during high aerosol loading condition according to the case study. I have no major concerns that need to be addressed. The manuscript is generally well written, clearly presented and is recommended for publication in AMT after some minor corrections.*

*Minor comments:*

*P1, L9: UV-> UV spectral region*

R: It has been corrected in the revised manuscript.

*P2, L27: Since no long-term aerosol extinction profile results were presented in the manuscript, it's better to describe with aerosol extinction properties, i.e. AOD, surface AEC.*

R: 'aerosol extinction profiles' -> 'aerosol extinction properties'.

*P3, L17&L22: The MAX-DOAS measurements are working in both UV and visible band, however, why did the authors only use the O4 absorptions in UV to retrieve the aerosol extinction profiles? Is it possible to show some inter-comparison of aerosol retrieval results between different wavelength bands?*

R: It's better to show the retrievals from different wavelength and related inter-comparison. However, we had a problem to record the spectra in the visible range. Unfortunately, no retrievals can be obtained in the visible band during this campaign.

*P6, L26: Why the sensitivity study of the a priori profile was tested with these four specific shapes? Is it restricted with the algorithm itself or any other reasons? Obviously, it is not realistic that Gaussian distribution as shown in Fig. 3(b).*

R: These four shapes of a priori are the default types of parameterization of a priori in the inversion algorithm. In principle, Gaussian distribution is completely differ from the others. Even though it is not realistic in a normal case, Gaussian distribution is more suitable to capture the elevated plume layer, e.g. desert dust plume. Please also refer to the response to Referee #2.

*P7, L31: The comparison of AODs from MAX-DOAS and AERONET is not so well under high aerosol loading situation. If the correlation between them was improved if the dust days was excluded from the statistics?*

R: Please refer to the revised Figure 4, in which we have supplemented a scatter plot of AODs from MAX-DOAS and AERONET.

*P8, L10: Please add one more sentence or proper reference for the estimation of light path based on O4 SCD at horizontal direction.*

R: More details about the approximation method can be found in Sinreich et al., 2013 and Wang et al., 2014. We have referred these two papers in the revised manuscript.

*P9, L32: Figs. ->Figure*

R: It has been corrected in the revised manuscript.

*Figure 3: delete 'vertical'*

R: We have corrected it.

*Figure 6: correlation plots -> linear regression plots*

R: 'correlation plots' have been replaced with 'linear regression plots'

Sinreich, R., Merten, A., Molina, L., and Volkamer, R.: Parameterizing radiative transfer to convert MAX-DOAS dSCDs into near-surface box-averaged mixing ratios, Atmos. Meas. Tech., 6, 1521-1532, doi: 10.5194/amt-6-1521-2013, 2013.

Wang, Y., Li, A., Xie, P. H., Wagner, T., Chen, H., Liu, W. Q., and Liu, J. G.: A rapid method to derive horizontal distributions of trace gases and aerosols near the surface using multi-axis differential optical absorption spectroscopy, Atmos. Meas. Tech., 7, 1663-1680, doi:10.5194/amt-7-1663-2014, 2014.